# Switchable Na⁺ and K⁺ selectivity in an amino acid functionalized 2D covalent organic framework membrane

Li Cao [1], I-Chun Chen[1], Zhen Li[1], Xiaowei Liu[1], Muhammad Mubashir[1], Reham Al Nuaimi[1] & Zhiping Lai [1]✉

Biological cell membranes can efficiently switch Na⁺/K⁺ selectivity in response to external stimuli, but achieving analogous functions in a single artificial membrane is challenging. Here, we report highly crystalline covalent organic framework (COF) membranes with well-defined nanochannels and coordinative sites (i. e., amino acid) that act as ion-selective switches to manipulate Na⁺ and K⁺ transport. The ion selectivity of the COF membrane is dynamic and can be switched between K⁺-selective and Na⁺-selective in a single membrane by applying a pH stimulus. The experimental results combined with molecular dynamics simulations reveal that the switchable Na⁺/K⁺ selectivity originates from the differentiated coordination interactions between ions and amino acids. Benefiting from the switchable Na⁺/K⁺ selectivity, we further demonstrate the membrane potential switches by varying electrolyte pH, miming the membrane polarity reversal during neural signal transduction in vivo, suggesting the great potential of these membranes for in vitro biomimetic applications.

Biological cell membranes enable efficient and selective ion transport, which plays a key role in most physiological processes[1–3]. For example, during neural signal transduction, external stimuli trigger the selective influx of Na⁺ and efflux of K⁺, causing a reverse in the membrane potential and thereby the nerve impulse transmission[4,5]. Fabricating artificial membranes with analogous functions (i.e., responsiveness to external stimuli and switchable Na⁺/K⁺ selectivity) has profound implications for fundamentally understanding the ion transport mechanisms in biological ion channels and for energy-efficient separation applications. However, as Na⁺ and K⁺ both are monovalent, achieving efficient Na⁺/K⁺ selectivity in artificial membranes remains a daunting challenge.

Conventional nanoporous membranes made of two-dimensional laminates[6–10] and polymers[11–18] usually exhibit poor mono/monovalent cations selectivity due to their broad channel size distribution. Metal-organic frameworks (MOFs)[19–23] and porous organic cages[24] possess ordered channels, but the weak interactions between monovalent cations and channel walls[18,25,26] compromise the selectivity. Macrocyclic

molecules, such as crown ethers[27,28], cucurbituril[29], and calixarene[30], can bind specific monovalent cations (e. g., Na⁺ and K⁺), which produce remarkable mono/monovalent cations selectivity[25,27,31,32]. However, these materials are not responsive, and the Na⁺/K⁺ selectivity cannot be reversed in a single device or membrane. Recently, covalent organic frameworks (COFs) membranes have shown potential in ion sieving[33–36], such as cation/anion[37–40], and mono/multi-valent cations separations[41–43], benefiting from their well-defined structures, suitable pore sizes, and flexible functionalities[44–46]. However, switchable Na⁺/K⁺ selective COF membranes have not yet been achieved due to their uncontrollable interactions with monovalent ions. Amino acids, the fundamental building units in protein channels, can efficiently recognize and guide ion transport through specific binding and reversible coordination with target ions[4,47]. Therefore, we envision that amino acids could be potential ion-selective switches that enable switchable Na⁺/K⁺ selectivity in a single artificial membrane when they are anchored onto the ordered COF nanochannels.

[1]Division of Physical Science and Engineering, 4700 King Abdullah, University of Science and Technology (KAUST), Thuwal 23955-6900, Kingdom of Saudi Arabia. ✉e-mail: zhiping.lai@kaust.edu.sa

Here, we demonstrate switchable Na⁺/K⁺ selective membranes based on covalent organic frameworks (COFs). The COF membranes were designed to constitute ordered sub-3 nm channels that allow for efficient grafting of amino acids (e. g., cysteine) via a rapid thiol-ene click reaction, producing cysteine functionalized COF membranes (COF-Cys). The cysteine serves as an ion-selective switch that responds to external stimuli, such as pH. The ion diffusion results revealed that the introduction of cysteine favored the transport of Na⁺ over K⁺, eventually yielding a reverse in Na⁺/K⁺ selectivity. More importantly, these COF-Cys membranes are pH-responsive, and a single COF-Cys-60% membrane can be switched between K⁺-selective (pH=3.8) and Na⁺-selective (pH=8.9) by applying a pH stimulus (Fig. 1), which is similar to the fundamental functions of biological cell membranes. We further demonstrate the membrane polarity switches by varying the solution pH.

## Results

### Synthesis and characterization of COF-Cys membranes

We first applied a multivariate strategy to incorporate vinyl groups into COFs (COF-V). The vinyl groups in COF-V allow for further post-synthetic functionalization via a rapid thiol-ene click reaction, thus producing cysteine-functionalized COFs (COF-Cys) (Fig. 2a). The multivariate strategy combined with click chemistry enables the simultaneous introduction of functional groups into the frameworks while maintaining the high crystallinity of COFs. The initial COF-V membranes were synthesized on polyacrylonitrile (PAN) support using an interfacial condensation reaction between 1,3,5-tri(4-aminophenyl) benzene (TAPB), 2,5-dimethoxyterephthalaldehyde (DMTP) and 2,5-divinylterephthalaldehyde (DVA). The resulting membranes were denoted as COF-V-x (x = 30%, 60%, and 80%), where x refers to the molar percentage of DVA over the total of DMTP and DVA. Here, we demonstrate the basic characterization results by taking the COF-V-60% membrane as an example. Scanning electron microscopy (SEM, Fig. 2b) revealed that the as-synthesized COF-V-60% membrane was uniform, intact, and defect-free. The thickness of the COF-V-60% membrane can be adjusted from 50 nm to 200 nm by varying the concentrations of monomers, as verified by the cross-section SEM images in Supplementary Fig. 1. The imine and vinyl structures of the resulting COF-V-60% membrane were confirmed by Fourier-transform infrared spectroscopy (FT-IR, Supplementary Fig. 2), solid-state ¹³C nuclear magnetic resonance (¹³C NMR, Supplementary Fig. 3), and X-ray photoelectron spectroscopy (XPS, Supplementary Fig. 4). The COF-V-60% membrane exhibited a type IV N₂ sorption isotherm, indicating its mesoporous character (Supplementary Fig. 5a). The Brunauer–Emmett–Teller (BET) surface area was estimated to be 810.7 m² g⁻¹ (Supplementary Fig. 5b). The pore size distribution profiles derived from the N₂ adsorption isotherms revealed a pore size of ~2.73 nm, which was in good agreement with the simulated structure (Supplementary Fig. 5c).

For all COF-V-x membranes, the grazing incidence wide-angle X-ray scattering (GIWAXS) patterns appeared as rings, indicating that the crystal domains were randomly oriented in the membranes (Fig. 2c, insets). The reflections from the GIWAXS patterns correspond to the 100, 110, 200, 210, and 220 lattice planes, which agree well with the powder X-ray diffraction pattern of the powder sample (Supplementary Fig. 6), suggesting high crystallinity of the synthesized COF-V-x membranes (Fig. 2c). The ordered pore structure was further visualized using low-dose high-resolution transmission electron microscopy (HR-TEM), as shown in Fig. 2e. Overall, the high crystallinity and porosity, combined with the large pore size, allow for the efficient introduction of functional groups into the nanochannels in COF-V-x membranes.

Considering the reversible coordination between amino acids and ions (e. g., K⁺ and Na⁺) in biological ion channels, an amino acid (i. e., cysteine) was then anchored onto the channel walls using a thiol-ene click reaction between vinyl groups and thiol groups in cysteine, initiated by UV light. The successful grafting of cysteine was confirmed by FT-IR, solid-state ¹³C NMR, and XPS analysis. For simplicity, we focus on the chemical structure characterization of the COF-Cys-60% membrane. FT-IR spectrum of the COF-Cys-60% membrane exhibited the characteristic bands of COOH and NH- at 1632 and 1531 cm⁻¹, respectively, suggesting that the cysteine was grafted onto the frameworks (Supplementary Fig. 7). This was further verified via ¹³C NMR, which exhibited new peaks at 30.1 ppm and 34.2 ppm, corresponding to alkyl C from cysteine molecules (Supplementary Fig. 8), and via XPS spectrum showing the sulfur signal at 168.3 eV (Supplementary Fig. 9). Notably, this rapid click reaction enabled the efficient incorporation of cysteine into the frameworks while the resulting COF-Cys-x membranes maintained high crystallinity and ordered channel structure. As shown in Fig. 2d, all COF-Cys-x membranes remained crystalline, showing identical GIWAXS patterns to COF-V-x membranes, which also suggested the lattice structures of COF-Cys-x membranes were not affected by the cysteine on channel walls. The HR-TEM image also revealed that the COF-Cys-60% membrane maintained ordered channel structures after post-modification (Fig. 2f). The introduction of cysteine on channel walls led to a decrease in both surface area and channel size. Specifically, the BET surface area decreased to 683.5, 648.5, and 504.8 m² g⁻¹ for COF-Cys-30%, COF-Cys-60%, and COF-Cys-80% membranes, respectively (Supplementary Fig. 10). Their pore sizes were 2.55, 2.52, and 2.51 nm, respectively, slightly smaller than

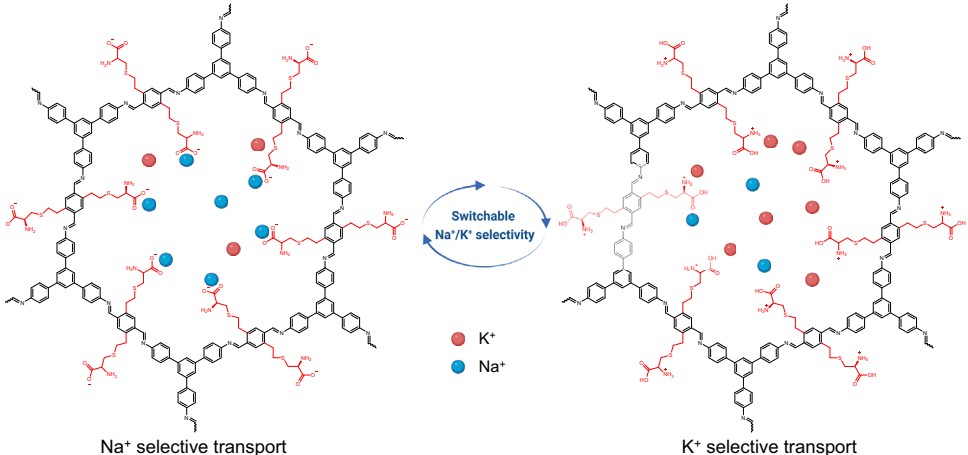

**Fig. 1 | Switchable Na⁺/K⁺ selectivity.** Switchable Na⁺/K⁺ selectivity enabled by a cysteine functionalized COF membrane. Red and blue balls refer to K⁺ and Na⁺, respectively.

Na⁺ selective transport

K⁺ selective transport

K⁺
Na⁺

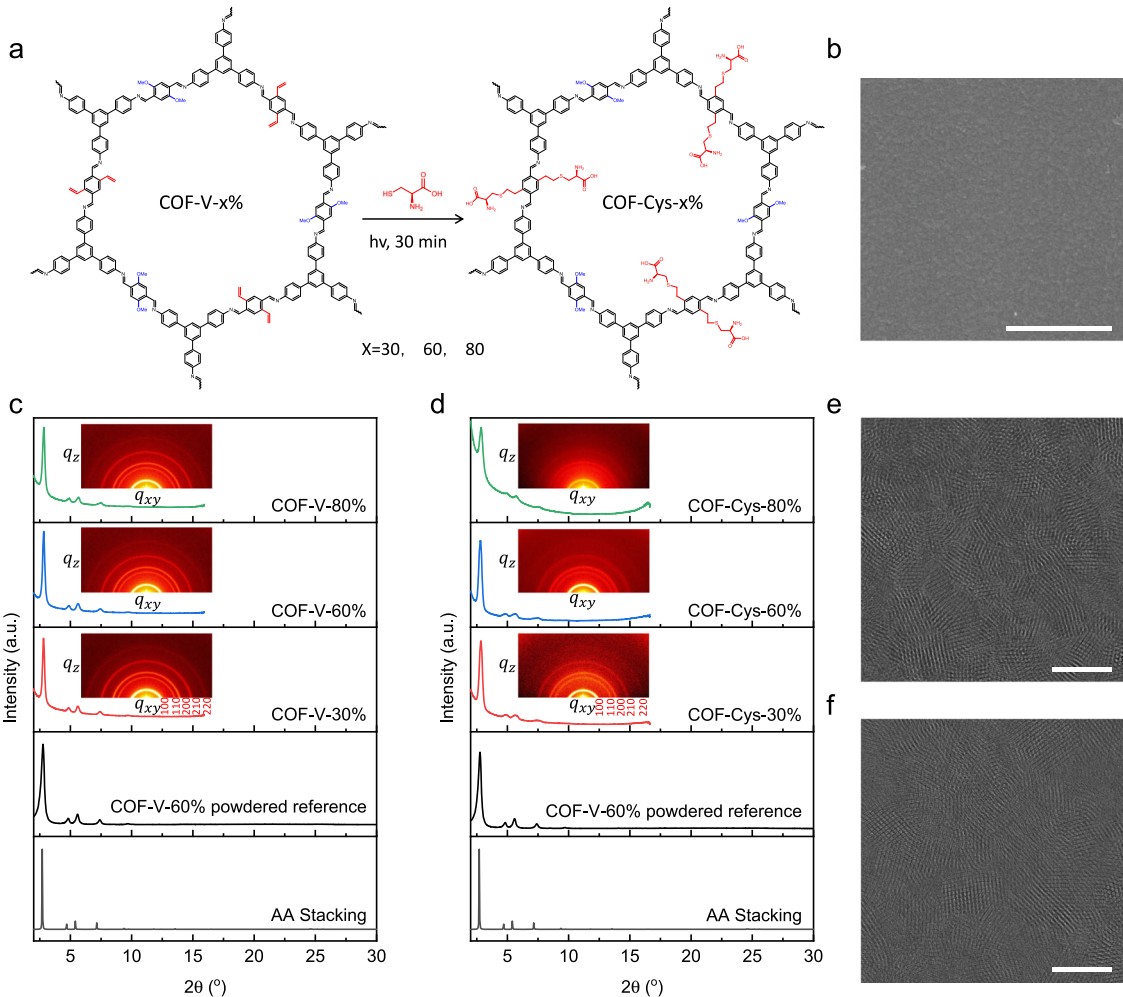

**Fig. 2 | Preparation and characterization of COF-Cys membranes. a** Interfacial polymerization of COF-V-x membranes and rapid synthesis of COF-Cys-x membranes via a thiol-ene click reaction. **b** SEM image of COF-Cys-60% membrane showing defect-free surface morphology (scale bar, 500 nm). Projection of the in-plane diffraction patterns from the grazing incidence wide-angle X-ray scattering (GIWAXS) data: **c** COF-V-x membranes, **d** COF-Cys-x membranes on PAN support (x = 30%, 60%, and 80%). Insets show GIWAXS patterns. The PXRD pattern of COF-V-60% powder is included as a reference. Low-dose motion-corrected high-resolution TEM (HRTEM) images: **e** COF-V-60% membrane (scale bar, 50 nm), **f** COF-Cys-60% membrane (scale bar, 50 nm), suggesting high crystallinity.

their parent COF-V-x membranes (Supplementary Fig. 11). SEM images indicated that the functionalized membranes possessed similar morphology and thickness to the pristine membrane (Supplementary Fig. 12).

## Switchable Na⁺/K⁺ selectivity by introducing cysteine into Sub-3 nm nanochannels

The ion transport behavior through COF membranes was investigated using a concentration-driven diffusion test. Ion diffusion fluxes were first measured with single-salt solutions, including 0.1 M NaCl or 0.1 M KCl. As shown in Supplementary Fig. 13, the ion concentration on the permeate side increased linearly with the diffusion time for all tested membranes. The calculated permeation rates of Na⁺ and K⁺ for the COF-V-60% membrane were 9.1 and 16.2 mmol m⁻² h⁻¹, respectively (Supplementary Fig. 14a), yielding an ideal K⁺/Na⁺ selectivity of 1.8 (Fig. 3a) which was close to the ratio of their diffusion coefficients in the bulk solution ($1.33 \times 10^{-9}\,m^2\,s^{-1}$ and $1.96 \times 10^{-9}\,m^2\,s^{-1}$ for Na⁺ and K⁺, respectively). The diffusion coefficients of K⁺ and Na⁺ in the COF-V-60% membrane were estimated to be $6.75 \times 10^{-11}\,m^2\,s^{-1}$ and $3.79 \times 10^{-11}\,m^2\,s^{-1}$, respectively. The decreased transport rate in the COF-V-60% membrane indicates that the ion transport is affected by the confined channel size and surface chemistry. This result also suggests that the trend of ion diffusion (K⁺ transports faster than Na⁺) through the sub-3 nm

nanochannels of the COF-V-60% membrane is the same as their bulk solutions. The introduction of cysteine facilitated both Na⁺ and K⁺ fluxes, which can be attributed to the interactions between the partially dissociated carboxyl groups and ions (pH 7.1). The COF-Cys-30% membrane showed 1.5- and 1.2-fold Na⁺ and K⁺ fluxes compared to the pristine COF-V-60% membrane, respectively (Supplementary Fig. 14b). As a result, the COF-Cys-30% membrane exhibited an ideal K⁺/Na⁺ selectivity of 1.4, which was slightly smaller than that of the COF-V-60% membrane. The reduced K⁺/Na⁺ selectivity suggested that the cysteine molecules in COF nanochannels favored the transport of Na⁺ over K⁺. As only a limited fraction (30%) of the nanochannels were functionalized with cysteine, the overall ion selectivity of the COF-Cys-30% membrane was governed by the nature of the pristine nanochannels in the COF-V-60% membrane. Therefore, the COF-Cys-30% membrane still exhibited the K⁺-selective characteristic.

If the cysteine could preferentially facilitate Na⁺ transport over K⁺, we anticipate that the ion selectivity might be reversed when more cysteine was incorporated into the nanochannels of COF-V membranes. The effects of cysteine on ion transport were then investigated by increasing the cysteine loading from 30% to 80%. Similarly, the permeation rates of both Na⁺ and K⁺ increased with cysteine loading. In particular, the permeation rate of Na⁺ increased sharply to 61.3 mmol h⁻¹ m⁻² for the COF-Cys-60% membrane

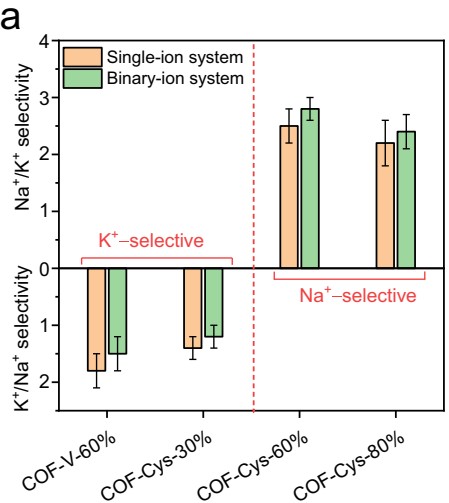
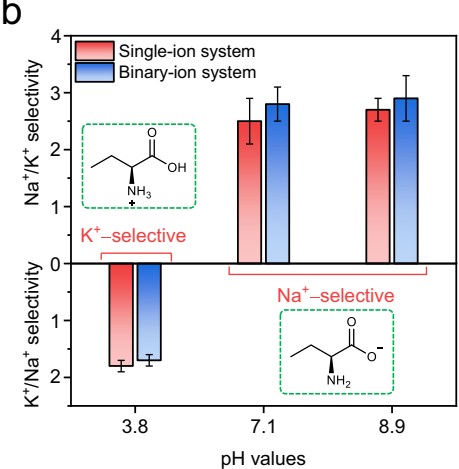

**Fig. 3 | Switchable Ion selectivity. a** Na$^+$/K$^+$ selectivity for COF-V-60% membrane and COF-Cys membranes with various cysteine loadings. **b** Influence of pH on Na$^+$/K$^+$ selectivity for COF-Cys-60% membrane. Error bars in **a** and **b** represent the standard deviation for three independent tests.

(Supplementary Fig. 14c). In contrast, the permeation rate of K$^+$ slightly increased to 24.5 mmol h$^{-1}$ m$^{-2}$. As a result, the COF-Cys-60% membrane exhibited reverse ion selectivity against the COF-Cys-30% membrane, favoring Na$^+$ transport with an ideal Na$^+$/K$^+$ selectivity of 2.5. However, the permeation rate of both Na$^+$ and K$^+$ for COF-Cys-80% was quite close to that of the COF-Cys-60% membrane (Supplementary Fig. 14d). This was mainly attributed to the partial collapse of the ordered channel structures in the COF-Cys-80% membrane, as revealed by the GIWAXS and N$_2$ sorption results (Fig. 2d and Supplementary Fig. 10c), highlighting the importance of the ordered nanochannels for ion sieving.

The single-ion diffusion results suggested that the reverse in ion selectivity originated from the substantially enhanced Na$^+$ flux. Evidently, the incorporation of cysteine in COF nanochannels preferentially facilitated Na$^+$ diffusion, and this facilitated transport effect was more pronounced in the binary-ion systems. As shown in Fig. 3a (green column), the binary-ion diffusion results revealed that the Na$^+$/K$^+$ selectivity was 2.8 for the COF-Cys-60% membrane measured in the binary-ion system, which was 17% higher than that obtained in the single-ion system (Supplementary Fig. 15). In general, the competition between cations in a binary-ion system will compromise the ion selectivity for most nanoporous membranes with sub-nanometer channels[19,24]. Instead, the enhanced Na$^+$/K$^+$ selectivity of the COF-Cys-60% membrane in the binary-ion system is similar to that observed in the polymeric membranes with specific binding sites, which suggests that a facilitated transport mechanism may dominate the ion transport process in the COF-Cys-60% membranes[16,17,48]. Overall, these results demonstrated that COF membranes provided a powerful platform to manipulate the monovalent cation transport by introducing cysteine into the ordered nanochannels to achieve switchable Na$^+$/K$^+$ selectivity.

## pH-switched Na$^+$/K$^+$ selectivity in a single COF membrane

Given that the residues in cysteine contain pH-sensitive carboxyl and amino groups, we next investigated how pH affects ion transport across the COF-Cys-60% membrane. Since the isoelectric point for cysteine is 5.02, we thus investigated the ion diffusion properties at two representative pH conditions, that is, pH 3.8 and pH 8.9. At pH 3.8, the permeation rate of Na$^+$ and K$^+$ for the COF-Cys-60% membrane was 12.6 and 21.4 mmol h$^{-1}$ m$^{-2}$, respectively (Supplementary Fig. 16a), and hence the COF-Cys-60% membrane exhibited K$^+$-selective transport with an ideal K$^+$/Na$^+$ selectivity of 1.7 (Fig. 3b). By contrast, when the solution pH was increased to 8.9, the COF-Cys-60% membrane presented a substantially enhanced Na$^+$ flux of 73.7 mmol h$^{-1}$ m$^{-2}$ while

maintaining the K$^+$ flux of 27.3 mmol h$^{-1}$ m$^{-2}$ (Supplementary Fig. 16b). Consequently, the enhanced Na$^+$ flux caused a reversed ion selectivity of the COF-Cys-60% membrane with a Na$^+$/K$^+$ selectivity of 2.7. More importantly, the same ion transport behavior was found in a binary-ion system with a slightly higher Na$^+$/K$^+$ selectivity of 2.9 for pH 8.9. In such a way, a single COF-Cys-60% membrane can be switched between Na$^+$-selective and K$^+$-selective by simply adjusting the solution pH. It should be noted that the pH-switched Na$^+$/K$^+$ selectivity of the COF-Cys-60% membrane is uncharacteristic of conventional stimuli-responsive membranes (e. g., pH-responsive membranes[13,16,49], and photo-responsive membranes[50,51]) that allow or reject all ions without preferential recognition of the target ion; that is, the conventional membranes can be switched between the ion-conducting and ion-insulating state using certain stimuli.

## pH-switched ion transport mechanisms

The switchable Na$^+$/K$^+$ selectivity achieved in a single COF-Cys-60% membrane by pH stimuli is unique and has not been observed in other artificial membranes. We propose that the cysteine molecules can be regarded as ion-selective switches manipulating Na$^+$ and K$^+$ transport by reversible coordination interactions (Fig. 4a). More specifically, no specific interactions occur between target ions and COF nanochannels at pH 3.8. The channel size of the COF-Cys-60% membrane is approximately 2.5 nm, which allows both Na$^+$ and K$^+$ to pass through freely. Considering that K$^+$ (8.92 × 10$^{-11}$ m$^{-2}$ s$^{-1}$) has a larger diffusion coefficient than Na$^+$ (5.25 × 10$^{-11}$ m$^{-2}$ s$^{-1}$), a higher K$^+$ flux is expected. Consequently, the COF-Cys-60% membrane yields a K$^+$/Na$^+$ selectivity of ~1.7. The reverse of ion selectivity (from K$^+$-selective to Na$^+$-selective) originates from the dramatically enhanced Na$^+$ flux at pH 8.9. Increasing the solution pH might activate the ion-selective switches, providing more binding sites for interacting with Na$^+$ and K$^+$. Here, these binding sites can be recognized as a catalyst that reduces the energy barrier of Na$^+$ permeation[26]. Therefore, the Na$^+$ flux was substantially enhanced, and eventually the COF-Cys-60% membrane favored Na$^+$ passage over K$^+$, resulting in a Na$^+$/K$^+$ selectivity of 2.7 at pH 8.9. We verified that altering solution pH cannot produce a reverse of Na$^+$/K$^+$ selectivity for the COF-V-60% membrane (Supplementary Fig. 17), implying that the reversible coordination interactions between target ions and cysteine molecules contribute to the switchable Na$^+$/K$^+$ selectivity in the COF-Cys-60% membrane.

To gain further insight into the ion transport mechanism in the COF-Cys-60% membrane at various pH conditions, the radial distribution function (RDF) was used to analyze the interactions between

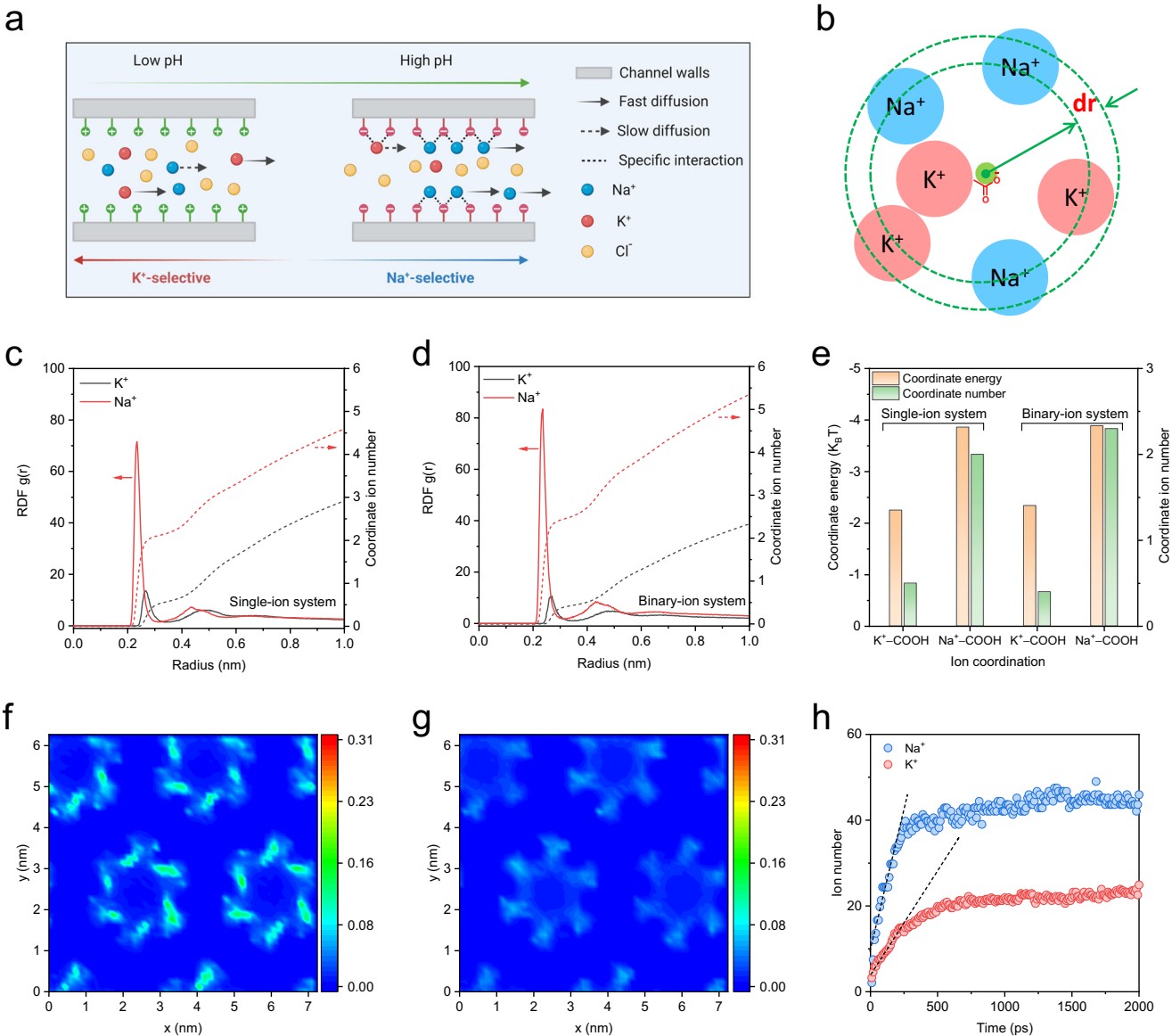

**Fig. 4 | Molecular dynamics simulation of ion transport. a** Schematics of the ion transport mechanism in cysteine functionalized nanochannels under various pH conditions. **b** Radial distribution function (RDF) of Na$^+$ and K$^+$ around cysteine in nanochannels. The cysteine is surrounded by Na$^+$ and K$^+$ and forms the ion shell. RDF of Na$^+$ and K$^+$ around COO$^-$ for COF-Cys-COO$^-$ membrane: **c** Singly-ion system; **d** Binary-ion system. **e** The coordinate number and energy of Na$^+$ and K$^+$ around COO$^-$ groups in COF-Cys-COO$^-$ membrane. **f** Density distribution profile of Na$^+$ in the xy-plane for COF-Cys-COO$^-$ membrane in a binary-ion system (1 M NaCl +1 M KCl). **g** Density distribution profile of K$^+$ in the xy-plane for COF-Cys-COO$^-$ membrane in a binary-ion system (1 M NaCl + 1 M KCl). **h** Numbers of Na$^+$ and K$^+$ transferred through COF-Cys-COO$^-$ membrane in binary-ion system plotted as functions of simulation time.

target ions and cysteine molecules (Fig. 4b). The fully dissociated carboxyl groups (denoted as COF-Cys-COO$^-$ for simulation) and protonated amino/carboxyl groups (denoted as COF-Cys-NH$_3^+$ for simulation) were used to simulate the cysteine molecules at pH 8.9 and pH 3.8 conditions, respectively. In the COF-Cys-COO$^-$ membrane nanochannels, the COO$^-$ groups are surrounded by both Na$^+$ and K$^+$ and form two ion shells. Here, the first ion shell of the COO$^-$ groups was analyzed to investigate the coordination properties between the target ions and COO$^-$ groups. As shown in Fig. 4c, the narrow and intensive peak of RDF (Na$^+$-COO$^-$) indicated that concentrated Na$^+$ was coordinated to the COO$^-$ groups, while K$^+$ exhibited broad and less intensive RDF (K$^+$-COO$^-$), suggesting loose coordination between K$^+$ and COO$^-$. A similar trend was found in the binary-ion system (Fig. 4d). We further calculated the coordinate ion number (Fig. 4c, d, dashed lines) and coordinate energy based on the RDF results to quantify the coordinate conditions. As exhibited in Fig. 4e, the coordinate Na$^+$ and K$^+$ numbers

of the COO$^-$ groups in the first layer were estimated to be 2 and 0.5, respectively. This result suggested that Na$^+$ was easier to coordinate with COO$^-$, which is further supported by its coordinate energy (−3.86 and −2.25 K$_B$T for Na$^+$ and K$^+$, respectively). In the binary-ion system, the coordinate energy for Na$^+$-COO$^-$ and K$^+$-COO$^-$ was −3.89 and −2.34 K$_B$T, respectively. The high affinity between Na$^+$ and COO$^-$ allowed Na$^+$ to enter the channels more easily and subsequently diffuse through a facilitated transport mechanism[17,48], and thus the Na$^+$ flux was significantly enhanced. The ion RDF around protonated amino (NH$_3^+$) and carboxyl groups (COOH) revealed weak coordination between ions (both for Na$^+$ and K$^+$) and NH$_3^+$/COOH (Supplementary Figs. 18, 19). This result indicated negligible interactions between ions and cysteine molecules[17], which explained the ion transport trend in the COF-Cys-60% membrane under solution pH 3.8.

Molecular dynamics (MD) simulations were performed to understand the ion transport process across the COF-Cys-60% membranes.

The simulation box consisted of a five-layer COF-Cys-COO⁻ membrane (corresponding to the COF-Cys-60% membrane at pH 8.9) or COF-Cys-NH₃⁺ membrane (corresponding to the COF-Cys-60% membrane at pH 3.8), 1 M KCl + 1 M NaCl mixture, and water molecules (Supplementary Fig. 20). We first calculated the potential mean force (PMF) profiles along the z-axis for both Na⁺ and K⁺. The energy barriers for Na⁺ and K⁺ to pass through the COF-Cys-COO⁻ membrane were found to be −2.7 and −1.4 $K_BT$, respectively (Supplementary Fig. 21), which indicated that the COF-Cys-COO⁻ membrane was more permeable to Na⁺. In contrast, both Na⁺ and K⁺ have to overcome an energy barrier when they pass through the positively charged COF-Cys-NH₃⁺ membrane. As shown in Supplementary Fig. 22, the energy barrier of Na⁺ (1.9 $K_BT$) passing through the COF-Cys-NH₃⁺ membrane was 1.3 times higher than that of K⁺ (1.5 $K_BT$).

We further calculated the probability distributions of Na⁺ and K⁺ in the xy-plane to visualize their transport process across the COF-Cys-COO⁻ and COF-Cys-NH₃⁺ membranes. As shown in Fig. 4f, g, both Na⁺ and K⁺ were concentrated around the COO⁻ groups in the COF-Cys-COO⁻ membrane due to the coordinative interactions between ions and COO⁻ groups on channel walls. Meanwhile, a higher density of Na⁺ was found in the COF-Cys-COO⁻ membrane (Fig. 4f). Closer quantitative analysis revealed that the Na⁺ and K⁺ concentrations were 0.49 and 0.25 M in the confined COF-Cys-COO⁻ nanochannels, respectively. By counting the total ion numbers passing through the COF-Cys-COO⁻ membrane, the time-dependent Na⁺ and K⁺ numbers were then plotted in Fig. 4h. The Na⁺ and K⁺ fluxes across the COF-Cys-COO⁻ membrane were calculated to be 138.3 and 60.5 ns⁻¹, respectively. The higher Na⁺ concentration in nanochannels and the higher Na⁺ flux explain the experimentally observed Na⁺ selectivity in the COF-Cys-60% membrane at pH 8.9. For the COF-Cys-NH₃⁺ membrane, both Na⁺ and K⁺ were randomly distributed in nanochannels (Supplementary Fig. 23). More importantly, the density of Na⁺ and K⁺ in COF-Cys-NH₃⁺ nanochannels was much lower than in COF-Cys-COO⁻ membrane. In summary, the simulation results highlight that the switchable Na⁺/K⁺ selectivity of the COF-Cys-60% membrane stems from the differentiated coordination interactions between ions and cysteine.

**Demonstration of membrane potential switches under stimuli**
Finally, we demonstrated the switchable membrane potential by reversing Na⁺/K⁺ selectivity in a single COF-Cys-60% membrane under various pH conditions. Fig 5a shows neural signal transduction by controlling the influx of Na⁺ and efflux of K⁺ across the biological cell membranes. Here, as a proof-of-concept, we modeled the membrane potential switching process in vitro by manipulating Na⁺ and K⁺ transport across a single COF-Cys-60% membrane. As shown in Fig. 5b, a COF-Cys-60% membrane was sandwiched between two chambers. The extracellular chamber was filled with 0.1 M NaCl and 5 mM KCl, and the intracellular chamber was filled with 0.1 M KCl and 5 mM NaCl. A pair of Ag/AgCl electrodes were used to measure the membrane potential. The chemical gradient drives Na⁺ migrating from the extracellular chamber to the intracellular chamber, while K⁺ migrates from the intracellular chamber to the extracellular chamber. As shown in Fig. 5c, the resting COF-Cys-60% membrane (pH=3.8) favored the transport of K⁺ over Na⁺, thereby generating a net efflux of K⁺ (from the intracellular chamber to the extracellular chamber) and thus a stable membrane potential of ~ -8.5 mV (Supplementary Fig. 24a, b, and Supplementary Fig. 25a, b). When a stimulus (pH 8.9) was applied to the COF-Cys-60% membrane, the Na⁺ facilitated transport sites were activated, which favored the transport of Na⁺ over K⁺. The net influx of Na⁺ (from the extracellular chamber to the intracellular chamber) changed the chemical gradient and in turn caused the membrane polarity to reverse, showing a membrane potential of ~+13.7 mV (Supplementary Fig. 24c, d and Supplementary Fig. 25c, d). More importantly, the membrane potentials were highly stable after five cycles by successively switching solution pH. The stable performance

was attributed to the structural integrity of the COF-Cys-60% membrane, as revealed by the GIWAXS and HR-TEM results (Supplementary Fig. 26). For comparison, the membrane potentials of the COF-V-60% membrane were also evaluated. As shown in Supplementary Fig. 27, it exhibited a membrane potential value of ~ −9.4 mV at pH 3.8, and this value slightly dropped to ~−8.8 mV when the solution pH was changed to 8.9. The result was reasonable and correlated to the constant K⁺/Na⁺ selectivity of the COF-V-60% membrane at various pH conditions. This excellent correlation also highlights the critical role of the switchable Na⁺/K⁺ selectivity of the COF-Cys-60% membrane in manipulating the membrane polarity.

## Discussion
Biological cell membranes have separate Na⁺ channels[47,52] and K⁺ channels[4,53], which contain perfectly arranged amino acids that can efficiently guide Na⁺ or K⁺ transport across the membrane. However, previous studies have faced challenges in controlling ion channel properties (e. g., channel size and chemical compositions) as perfectly as those of biological ion channels[18,25,26]. In this work, we turn to an alternative route to manipulate Na⁺ and K⁺ transport by installing ion-selective switches within 2.5-nm nanochannels of COF membranes. We exploited the intrinsic ion diffusion properties and the differentiated coordination interactions between ions and amino acids to achieve switchable Na⁺/K⁺ selectivity in a single COF membrane under pH stimuli. This work offers a new strategy for creating ion-selective membranes for precise ion or molecular sieving.

In summary, we have developed a powerful platform based on covalent organic frameworks (COFs) to manipulate Na⁺ and K⁺ transport. An amino acid (i. e., cysteine), known as the basic building units of biologic ion channels, was anchored on the channel walls to serve as the ion-selective switches. The cysteine-functionalized COF membranes were pH-sensitive, allowing for switchable Na⁺/K⁺ selectivity in a single membrane by altering solution pH. Specifically, the COF-Cys-60% membrane exhibited K⁺/Na⁺ selectivity of 1.7 at pH 3.8 and Na⁺/K⁺ selectivity of 2.9 at pH 8.9. Combined experimental results and molecular dynamic simulations revealed that the differentiated coordination interactions between ions and nanochannels contributed to the switchable Na⁺/K⁺ selectivity at various pH conditions. More importantly, the membrane polarity could be switched between the positive and negative states by controlling the influx of Na⁺ and efflux of K⁺, which mimicked the reverse in membrane potentials during neural signal transduction in vivo. Although the present COF membrane with switchable Na⁺/K⁺ selectivity cannot feature direct biological analogs, it provides further insight into the effect of local chemical compositions on ion transport, which is known to underlie the fundamental functions of biological ion channels. Our findings also provide opportunities to develop smart membranes for potential applications, including stimuli-gated ionic transistors, nanofluidic circuits, as well as precise ion separations.

## Methods
### Materials
1,3,5-tris(4-aminophenyl)-benzene (TAPB), 2,5-Dimethoxyterephthalaldehyde (PDA-OMe), 2,5-divinylterephthalaldehyde (PDA-V) were purchased from Jilin Chinese Academy of Sciences Yanshen Technology Co., Ltd, China. Ethyl acetate, mesitylene, acetic acid, ethanol, tetrahydrofuran (THF), dimethylformamide, potassium chloride, sodium chloride, hydrochloric acid, sodium hydroxide, potassium hydroxide, L-cysteine, and 2-Hydroxy-4'-(2-hydroxyethoxy) −2-methylpropiophenone (H1361) were purchased from Sigma Aldrich and used without further purification. Polyacrylonitrile (PAN) ultra-filtration membrane support with a molecular weight cut-off of 20 k PEG was supplied by Sepro Membranes Inc. (USA). The deionized water (18.2 MΩ·cm) was produced through a Millipore Milli-Q water purification system.

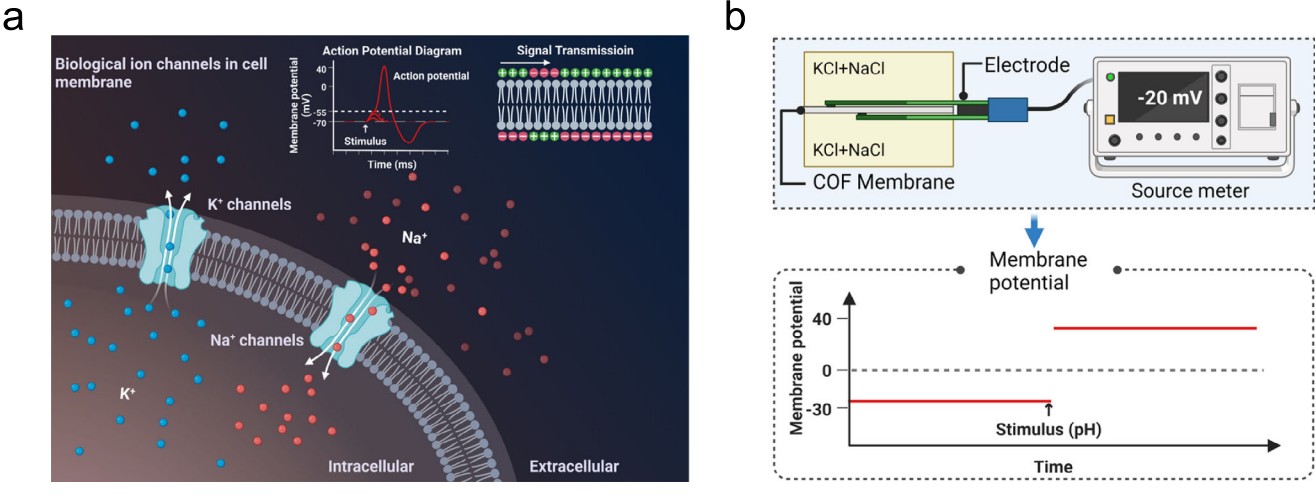

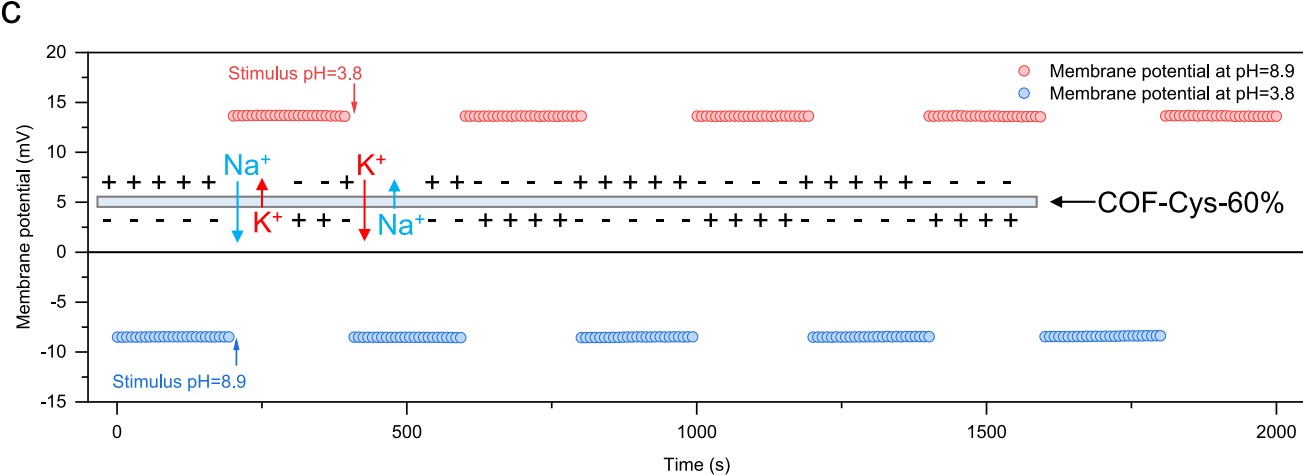

**Fig. 5 | Membrane potential switches under stimuli. a** A diagram showing Na⁺/K⁺ transport across the neuron membrane when stimulated to achieve signal transmission (created with BioRender.com). **b** Schematic illustration of membrane potential measurement (created with BioRender.com). A COF-Cys-60% membrane was sandwiched between an intracellular chamber (filled with 0.1 M KCl and 5 mM NaCl) and an extracellular chamber (filled with 0.1 M NaCl and 5 mM KCl). The membrane potential was recorded using a source meter. **c** Membrane potential switches when a stimulus is applied (pH).

## Fabrication of COF-V-x% membranes on PAN support

**COF-V-60% membrane.** COF-V-60% membranes were in-situ grown on a PAN ultrafiltration membrane support by using an interfacial synthesis method. Typically, a PAN membrane was placed between two chambers of a homemade diffusion cell with a volume of around 10 mL each. An aqueous solution of TAPB (0.025 mM in 10 mL 6 M acetic acid) and an organic solution of PDA-OMe/PDA-V (0.0225 mM of PDA-V and 0.015 mM of PDA-OMe in 10 mL ethyl acetate/mesytilene mixture (v/v: 4/1)) were introduced into two chambers. The whole system was kept at 30 °C for 4 days without disturb. The COF-V-60% membrane was uniformly grown on the PAN surface after 4 days and then was rinsed with ethyl acetate (3 × 20 mL), mesytilene (3 × 20 mL), tetrahydrofuran (3 × 20 mL) and ethanol (3 × 20 mL) to remove the unreacted monomers. Finally, the COF-V-60% membrane was dried at 50 °C under vacuum for 24 h. The COF-V-30% and COF-V-80% membranes were fabricated using a similar synthesis procedure to the COF-V-60% membrane (see Supplementary Methods).

## Fabrication of COF-Cys-x% membranes on PAN support

**COF-Cys-60% membrane.** A piece of COF-V-60% membrane was immersed into a 50 mL aqueous solution containing 121 mg of L-cysteine and 112 mg of H1361 initiator. The reaction mixture was then transferred to an UV crosslinker (XL–1000 UV CROSSLINKER, SPECTRO LINKER) and irradiated for 30 min at room temperature (20 °C ± 2 °C). The resulting COF-Cys-60% membrane was washed with deionized water (5 × 50 mL) to remove the residual L-cysteine and initiator. Finally, the COF-Cys-60% membrane was dried at 50 °C under vacuum for 24 h. The COF-Cys-30% and COF-Cys-80% membranes were fabricated using a similar synthesis procedure to the COF-Cys-60% membrane (see Supplementary Methods).

## Characterizations

Detailed Characterizations have been provided in the Supplementary Methods.

## Data availability

All data supporting the findings of this study are available within the article and the Supplementary Information file, or available from the corresponding authors upon request. Source data are provided with this paper.

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

## Acknowledgements

This work was supported by the KAUST Centre Competitive Fund FCC/1/1972–19 (Z. L.) and KAUST baseline fund BAS/A/1375-01 (Z. L.).

## Author contributions

L.C. and Z.L. conceived the idea. L.C. and I.C. fabricated the membranes. M.M., R.A. and X.L. did the basic characterization. L.C. and Z.Li. performed the electrochemical test and molecular dynamics simulations. L.C. and Z.L. wrote the manuscript. All authors discussed and revised the manuscript.

## Competing interests

The authors declare no competing interests.
