## [Peer Review File · Nature Communications]

Switchable Na⁺ and K⁺ Selectivity in an Amino Acid
Functionalized 2D Covalent Organic Framework MembraneReviewers' Comments:

Reviewer #1:

Remarks to the Author:

Please see the attached reviewer comments. A revision is required.

This manuscript reported a COF membrane with switchable Na^+/K^+ selectivity, achieved by applying a pH stimulus on the amino acid functionalized COF nanochannels. The dynamic Na^+/K^+ selectivity is ascribed to the differentiated coordination interactions between ions and coordinative sites of the nanochannels and demonstrated by the experiments and molecular dynamic simulations. Moreover, the membrane potential can be switched between the positive and negative states by controlling the influx of Na^+ and efflux of K^+ , mimicking the membrane polarity reverse during neural signal transduction in vivo. The novelty of this work is sufficient, and the experimental data and theoretical analysis are solid. Some specific comments are listed and should be addressed.

(1) “This result suggested that the ion transport behavior through the sub-3 nm nanochannels of the COF-V-60% membrane was comparable to that of their bulk solutions”. “That means the ion transport behavior across the COF-Cys-60% membrane are similar to that of their bulk solutions.” The authors claim the ion transport behaviors in the COF-V-60% membrane and the COF-Cys-60% membrane are similar to that of their bulk solutions, which is debatable. The confined channel size and channel surface chemistry should affect the transport behavior of K^+ and Na^+ . The ion diffusion coefficient of K^+ and Na^+ in the COF-V-60% and COF-Cys-60% channels can be calculated and give some evidence.

(2) “In all MD simulations, the simulation systems consisted ~ 67600 atoms with dimensions of approximately $7.24 \times 6.27 \times 15.0 \text{ nm}^3$, which were constructed by a graphene wall, a 5-layer COF membrane.” In the MD simulations of the COF membrane, why was a graphene wall introduced?

(3) For the binary-ion system, the feed solution of the experiment is 0.1 M NaCl and 0.1 M KCl, while for the simulation is 1 M NaCl+1 M KCl. Thus, the driving force for ion diffusion is different. Can the authors add some explanations?

(4) The scheme shown in Figure 4b is confusing. The coordinate number of Na^+ and K^+ around COO^- for COF-Cys- COO^- membrane is unreasonable, since the channel size is only 2.5 nm and the hydrated Na^+ and K^+ diameters are around 7 Å.

(5) “the grazing incidence wide-angle X-ray scattering (GIWAXS) patterns appeared

as rings, indicating that the crystal domains were randomly oriented in membranes (Figure 2c, insets).” “The ordered pore structure was further visualized using low-dose high-resolution transmission electron microscopy (HR-TEM)” The two terms “randomly oriented” and “ordered pore structure” seem contradictory.

(6) As described in the Introduction, “the broad channel size distribution” is not suitable for metal-organic frameworks and porous organic cages.

Reviewer #2:

Remarks to the Author:

In this manuscript, the authors successfully synthesized a 2D COF membrane with vinyl groups, and then post-modified cysteine to the pore wall of COF via a rapid thiol-ene "click" reaction, so that the material can be used as an ion-selective switch to manipulate the transport of Na⁺ and K⁺. The ion transport mechanism of COF-Cys-60% membrane at different pH conditions was further explored. Overall, the work is important and interesting for researchers in the field. Therefore, this referee would like to recommend the publication of the manuscript after minor revision. Specific suggestions are as follows.

1. In this work, the ion transport performance and selectivity of COF membrane were tested at different pH. However, the structure of COF is easy to be damaged in strong acid and base environment. Therefore, it is recommended that the stability of COF used in this research should be tested at the corresponding pH to ensure the integrity of COF membrane during testing.
2. It is suggested that the lines of COF in Figures 1 and 2 should be bolded, and the gray background in Figure 2A should be removed, so that the structure of COF is clearer and easier for readers to read.
3. In a and d of Figure 2, 2,5-divinylterephthalaldehyde (DVA) accounts for 40% of the two-node ligand for the synthesis of COF, but only 30% of the unmodified COF membrane in c. Similarly, these two proportions appear several times in the manuscript, so what is the true proportion used in this work? Please keep the manuscript data consistent with the experimental content.
4. In this paper, only the XRD data of COF-V-60% is quoted, and the stacking mode of 2D COF corresponding to the data is not indicated. In addition, other proportions were also synthesized in this work. It is suggested to supplement the XRD patterns of COFs with different proportions to determine their structures.
5. In the caption of Figure 2, parentheses are used in part of the letters, but not in other places in the manuscript. It is recommended to revise them to keep consistent.

A point-by-point response to referees' comments

Reviewer 1 (Remarks to the Author):

This manuscript reported a COF membrane with switchable Na⁺/K⁺ selectivity, achieved by applying a pH stimulus on the amino acid functionalized COF nanochannels. The dynamic Na⁺/K⁺ selectivity is ascribed to the differentiated coordination interactions between ions and coordinative sites of the nanochannels and demonstrated by the experiments and molecular dynamic simulations. Moreover, the membrane potential can be switched between the positive and negative states by controlling the influx of Na⁺ and efflux of K⁺, mimicking the membrane polarity reverse during neural signal transduction in vivo. The novelty of this work is sufficient, and the experimental data and theoretical analysis are solid. Some specific comments are listed and should be addressed.

Response: We greatly appreciate the reviewer for the highly positive comment.

(1) “This result suggested that the ion transport behavior through the sub-3 nm nanochannels of the COF-V-60% membrane was comparable to that of their bulk solutions”. “That means the ion transport behavior across the COF-Cys-60% membrane are similar to that of their bulk solutions.” The authors claim the ion transport behaviors in the COF-V-60% membrane and the COF-Cys-60% membrane are similar to that of their bulk solutions, which is debatable. The confined channel size and channel surface chemistry should affect the transport behavior of K⁺ and Na⁺. The ion diffusion coefficient of K⁺ and Na⁺ in the COF-V-60% and COF-Cys-60% channels can be calculated and give some evidence.

Response: What we mean here is the trend of ion diffusion, i.e., whether K⁺ moves faster or slower than Na⁺, is the same as the bulk solution. We are sorry for the confusion, and we agree with the reviewer that the confined channel size and channel surface chemistry do affect the transport speed of K⁺ and Na⁺. As suggested by the reviewer, we calculated the diffusion coefficients of K⁺ and Na⁺ in the COF-V-60% and COF-Cys-60% membranes (pH=3.8). The estimated diffusion coefficients of K⁺ ($6.75 \times 10^{-11} \text{ m}^2 \text{ s}^{-1}$ and $8.92 \times 10^{-11} \text{ m}^2 \text{ s}^{-1}$ for COF-V-60% and COF-Cys-60%, respectively) and Na⁺ ($3.79 \times 10^{-11} \text{ m}^2 \text{ s}^{-1}$ and $5.25 \times 10^{-11} \text{ m}^2 \text{ s}^{-1}$ for COF-V-60% and COF-Cys-60%, respectively) in COF membranes was lower than that in their bulk solutions. The decreased transport rate indicates that the ion transport is affected by the confined channel size and surface

chemistry, but trend of diffusion, that is K^+ transports faster than Na^+ , is the same as the bulk solution. A clear description is provided in the revision.

Revised manuscript

The calculated permeation rates of Na^+ and K^+ for the COF-V-60% membrane were 9.1 and 16.2 $mmol\ m^{-2}\ h^{-1}$, respectively (Supplementary Fig. 14a), yielding an ideal K^+/Na^+ selectivity of 1.8 (Fig. 3a) which was close to the ratio of their diffusion coefficients in the bulk solution ($1.33 \times 10^{-9}\ m^{-2}\ s^{-1}$ and $1.96 \times 10^{-9}\ m^{-2}\ s^{-1}$ for Na^+ and K^+ , respectively). The diffusion coefficients of K^+ and Na^+ in the COF-V-60% membrane were estimated to be $6.75 \times 10^{-11}\ m^{-2}\ s^{-1}$ and $3.79 \times 10^{-11}\ m^{-2}\ s^{-1}$, respectively. The decreased transport rate in the COF-V-60% membrane indicates that the ion transport is affected by the confined channel size and surface chemistry. This result also suggests that the trend of ion diffusion (K^+ transports faster than Na^+) through the sub-3 nm nanochannels of the COF-V-60% membrane is the same as their bulk solutions. The introduction of cysteine facilitated both Na^+ and K^+ fluxes, which can be attributed to the interactions between the partially dissociated carboxyl groups and ions (pH 7.1).

More specifically, no specific interactions occur between target ions and COF nanochannels at pH 3.8. The channel size of the COF-Cys-60% membrane is approximately 2.5 nm, which allows both Na^+ and K^+ to pass through freely. Considering that K^+ ($8.92 \times 10^{-11}\ m^{-2}\ s^{-1}$) has a larger diffusion coefficient than Na^+ ($5.25 \times 10^{-11}\ m^{-2}\ s^{-1}$), a higher K^+ flux is expected. Consequently, the COF-Cys-60% membrane yields a K^+/Na^+ selectivity of ~ 1.7 . The reverse of ion selectivity (from K^+ -selective to Na^+ -selective) originates from the dramatically enhanced Na^+ flux at pH 8.9.

(2) “In all MD simulations, the simulation systems consisted ~ 67600 atoms with dimensions of approximately $7.24 \times 6.27 \times 15.0\ nm^3$, which were constructed by a graphene wall, a 5-layer COF membrane.” In the MD simulations of the COF membrane, why was a graphene wall introduced? Response: We used a technique called Periodic Boundary Conditions (PBCs) in the MD simulations to eliminate problems caused by finite boundaries. With PBCs, when a particle crosses the left boundary of the cuboid simulation box, it enters the simulation box from the right boundary of the simulation box. In this situation, a graphene wall is usually introduced into the simulation box at both end sides (Nature 536.7615 (2016): 197-200; J. Am. Chem. Soc. 144 (2022): 12400–

12409; Nano Energy 57 (2019): 783-790.). Therefore, in our work, the graphene wall was added to make sure that the COF membrane is the only pathway for ions transport between the left side and the right side of COF membrane. We added an explanation in the revision for the introduction of the graphene wall.

Revised manuscript

In all MD simulations, the simulation systems consisted ~67600 atoms with dimensions of approximately $7.24 \times 6.27 \times 15.0 \text{ nm}^3$, which were constructed by a graphene wall, a 5-layer COF membrane, ~20000 TIP3P water molecules⁸ and ions ((i) K^+/Cl^- 1000 mM; (ii) Na^+/Cl^- 1000 mM, and (iii) K^+/Cl^- 1000 mM and Na^+/Cl^- 1000 mM). Here, a graphene was introduced into the simulation box to eliminate problems caused by finite boundaries, so that the COF membrane is the only pathway for ions transport between the left side and the right side of COF membrane. Then 60 ns simulation was performed for each system. The first 30 ns was discarded in each simulation trajectory for thermodynamic equilibration, followed by a 30 ns of production run.

(3) For the binary-ion system, the feed solution of the experiment is 0.1 M NaCl and 0.1 M KCl, while for the simulation is 1 M NaCl+1 M KCl. Thus, the driving force for ion diffusion is different. Can the authors add some explanations?

Response: The concentration of K^+/Na^+ is 1 M NaCl +1 M KCl in MD simulation, which can be explained as follows: Firstly, a 5-layer COF membrane was used in the simulation, which contains 120 e charges ($-\text{COO}^-$). Therefore, there should be 120 Na^+/K^+ to keep the system electrically neutral. Secondly, the time scale of MD simulation trajectory is typically on the order of tens of nanoseconds. Thus, additional K^+/Na^+ should be available in the simulation box to study the ion transport behavior. That means the number of K^+/Na^+ should be more than 120 in the simulation box. Considering the volume of the simulation box, the minimum concentration of K^+/Na^+ should be 0.67 M. Therefore, for the binary-ion system, the feed solution was 1 M $\text{Na}^+ + 1 \text{ M K}^+$. The above explanation has been included in the revised Supporting Information.

Revised Supporting Information

In all MD simulations, the simulation systems consisted ~67600 atoms with dimensions of approximately $7.24 \times 6.27 \times 15.0 \text{ nm}^3$, which were constructed by a graphene wall, a 5-layer COF membrane, ~20000

TIP3P water molecules⁷ and ions ((i) K^+/Cl^- 1000 mM; (ii) Na^+/Cl^- 1000 mM, and (iii) K^+/Cl^- 1000 mM and Na^+/Cl^- 1000 mM). Then 60 ns simulation was performed for each system. The first 30 ns was discarded in each simulation trajectory for thermodynamic equilibration, followed by a 30 ns of production run.

The concentration of K^+/Na^+ is 1 M NaCl/1 M KCl for MD simulation, which can be explained as follows: Firstly, a 5-layer COF membrane was used in the simulation, which contains 120 e charges ($-\text{COO}^-$). Therefore, there should be 120 Na^+/K^+ to keep the system electrically neutral. Secondly, the time scale of MD simulation trajectories is typically on the order of tens of nanoseconds. Thus, additional K^+/Na^+ should be available in the simulation box to study the transport behavior. That means the number of K^+/Na^+ should be more than 120 in the simulation box. Considering the volume of the simulation box, the minimum concentration of K^+/Na^+ should be 0.67 M. Therefore, the feed solution was 1 M Na^+/K^+ for the MD simulation.

(4) The scheme shown in Figure 4b is confusing. The coordinate number of Na^+ and K^+ around COO^- for COF-Cys- COO^- membrane is unreasonable, since the channel size is only 2.5 nm and the hydrated Na^+ and K^+ diameters are around 7 Å.

Response: Thanks for the reviewer's suggestion. In the initial scheme in Figure 4b, we overlooked the size issue of the ions and the COF channels. We reduced the coordinate number of Na^+ and K^+ and the scheme has been redrawn in the revised manuscript (Figure 4b).

Revised manuscript

Figure 4. Molecular dynamics simulation of ion transport.

(5) “the grazing incidence wide-angle X-ray scattering (GIWAXS) patterns appeared as rings, indicating that the crystal domains were randomly oriented in membranes (Figure 2c, insets).” “The ordered pore structure was further visualized using low-dose high-resolution transmission electron microscopy (HR-TEM)” The two terms “randomly oriented” and “ordered pore structure” seem contradictory.

Response: The low-dose high-resolution transmission electron microscopy (HR-TEM) shows that the membrane is polycrystalline containing randomly oriented crystal domains. Each domain is highly crystalline with “ordered pore structure”. The grazing incidence wide-angle X-ray

scattering (GIWAXS) patterns appeared as uniform rings, which is the result of the randomly oriented crystal domains. Therefore, the crystal domains have “ordered pore structure” (Figure R1), and these crystal domains are “randomly oriented” in the bulk COF membranes (Figure R2).

Figure R1. The COF crystal domains have “ordered pore structure”.

Figure R2. The crystal domains were “randomly oriented” in the bulk COF membrane.

(6) As described in the Introduction, “the broad channel size distribution” is not suitable for metal-organic frameworks and porous organic cages.

Response: We agree with the reviewer that “the broad channel size distribution” is not suitable for metal-organic frameworks and porous organic cages. A clearer description has been included in the revised manuscript.

Revised manuscript

Introduction

Fabricating artificial membranes with analogous functions (i.e., responsiveness to external stimuli and switchable Na^+/K^+ selectivity) has profound implications for fundamentally understanding the ion transport mechanisms in biological ion channels and energy-efficient separation applications. However, as Na^+ and K^+ both are monovalent, achieving efficient Na^+/K^+ selectivity in artificial membranes remains a daunting challenge.

Conventional nanoporous membranes made of two-dimensional laminate⁶⁻¹⁰ and polymers¹¹⁻¹⁸ usually exhibited poor mono/monovalent cations selectivity due to the broad channel size distribution. Metal-organic frameworks (MOFs)¹⁹⁻²³ and porous organic cages²⁴ possess ordered channels, but the weak interactions between monovalent cations and channel walls^{18,25,26} compromise the selectivity. Macrocyclic molecules, such as crown ether^{27,28}, cucurbituril²⁹, and calixarene³⁰, can bind specific monovalent cations (e. g., Na^+ and K^+), which produces remarkable mono/monovalent cations selectivity^{25,27,31,32}.

Reviewer 2 (Remarks to the Author):

In this manuscript, the authors successfully synthesized a 2D COF membrane with vinyl groups, and then post-modified cysteine to the pore wall of COF via a rapid thiol-ene “click” reaction, so that the material can be used as an ion-selective switch to manipulate the transport of Na^+ and K^+ . The ion transport mechanism of COF-Cys-60% membrane at different pH conditions was further explored. Overall, the work is important and interesting for researchers in the field. Therefore, this referee would like to recommend the publication of the manuscript after minor revision. Specific suggestions are as follows.

Response: We greatly appreciate the reviewer for the highly positive comment.

1. In this work, the ion transport performance and selectivity of COF membrane were tested at different pH. However, the structure of COF is easy to be damaged in strong acid and base environment. Therefore, it is recommended that the stability of COF used in this research should be tested at the corresponding pH to ensure the integrity of COF membrane during testing.

Response: Thanks for the reviewer’s suggestion. In order to investigate the stability of COF membranes, GIWAXS and HR-TEM measurements were performed for the COF-Cys-60% membrane after five-cycle’s membrane potential test (successively switching solution pH from 3.8 to 8.9). As shown in Figure R1 and Figure S26 in the revised Supporting Information, both GIWAXS and HR-TEM results indicated that the COF-Cys-60% membrane maintained high crystallinity, thereby confirming the high stability of the COF-Cys-60% membrane.

Figure R1. a. GIWAXS data of COF-Cys-60% membrane after 5 cycles test. b. Projection of the in-plane diffraction patterns from the GIWAXS data. c. HR-TEM image of COF-Cys-60% membrane after 5 cycles test (scale bar, 50 nm).

Revised Supporting Information

Figure S26. a. GIWAXS data of COF-Cys-60% membrane after 5 cycles test. b. Projection of the in-plane diffraction patterns from the GIWAXS data. c. HR-TEM image of COF-Cys-60% membrane after 5 cycles test (scale bar, 50 nm).

2. It is suggested that the lines of COF in Figures 1 and 2 should be bolded, and the gray background in Figure 2A should be removed, so that the structure of COF is clearer and easier for readers to read.

Response: Thanks for the reviewer's suggestion. To make the structure of COF clearer and easier for reader to read, the line of COF in Figure 1 and 2 has been bolded and the gray background in Figure 2A has been removed in the revised manuscript.

Revised manuscript

Figure 1. Switchable Na⁺/K⁺ selectivity enabled by a cysteine functionalized COF membrane.

Figure 2. Preparation and characterization of COF-Cys membranes. **a.** Interfacial polymerization of COF-V-x membranes and rapid synthesis of COF-Cys-x membranes via a thiolene click reaction. **b.** SEM image of COF-Cys-60% membrane showing defect-free surface morphology (scale bar, 500 nm). Projection of the in-plane diffraction patterns from the grazing incidence wide-angle X-ray scattering (GIWAXS) data: **c.** COF-V-x membranes, **d.** COF-Cys-x membranes on PAN support (x=30%, 60%, and 80%). Insets show GIWAXS patterns. The PXRD pattern of COF-V-60% powder is included as a reference. Low-dose motion-corrected high-resolution TEM (HRTEM) images: **e.** COF-V-60% membrane (scale bar, 50 nm), **f.** COF-Cys-60% membrane (scale bar, 50 nm), suggesting high crystallinity.

3. In a and d of Figure 2, 2,5-divinylterephthalaldehyde (DVA) accounts for 40% of the two-node ligand for the synthesis of COF, but only 30% of the unmodified COF membrane in c. Similarly,

these two proportions appear several times in the manuscript, so what is the true proportion used in this work? Please keep the manuscript data consistent with the experimental content.

Response: We are sorry for the mistake. The proportion should be 30%. The whole manuscript has been carefully checked and the revised manuscript is consistent with the experimental content.

Revised manuscript

Figure 2. Preparation and characterization of COF-Cys membranes. **a.** Interfacial polymerization of COF-V-x membranes and rapid synthesis of COF-Cys-x membranes via a thiolene click reaction. **b.** SEM image of COF-Cys-60% membrane showing defect-free surface morphology (scale bar, 500 nm). Projection of the in-plane diffraction patterns from the grazing

incidence wide-angle X-ray scattering (GIWAXS) data: **c.** COF-V-x membranes, **d.** COF-Cys-x membranes on PAN support (x=30%, 60%, and 80%). Insets show GIWAXS patterns. The PXRD pattern of COF-V-60% powder is included as a reference. Low-dose motion-corrected high-resolution TEM (HRTEM) images: **e.** COF-V-60% membrane (scale bar, 50 nm), **f.** COF-Cys-60% membrane (scale bar, 50 nm), suggesting high crystallinity.

Results and discussion

Synthesis and Characterization of COF-Cys Membranes

The resulting membranes were denoted as COF-V-x (x=30%, 60%, and 80%), where x refers to the molar percentage of DVA over the total of DMTP and DVA.

4. In this paper, only the XRD data of COF-V-60% is quoted, and the stacking mode of 2D COF corresponding to the data is not indicated. In addition, other proportions were also synthesized in this work. It is suggested to supplement the XRD patterns of COFs with different proportions to determine their structures.

Response: Thanks for the reviewer's suggestion. As shown in Figure R1, the 2D COFs were stacked in the AA stacking mode, which were well in accordance with the previous work (J. Am. Chem. Soc. 2017, 139, 2786–2793). The AA stacking mode has been added in Figures 2c and d in the revised manuscript.

Figure R1. Projection of the in-plane diffraction patterns from the grazing incidence wide-angle X-ray scattering (GIWAXS) data of COF-V-x membranes on PAN support (x=30%, 60%, and 80%). The PXRD patterns of COF-V-60% powder and the simulated AA stacking mode are included as references.

The powder XRD patterns of COFs with different proportions (COF-V-30%, COF-V-60%, and COF-V-80%) were shown in Figure R2. These COFs materials exhibited similar XRD patterns with an AA stacking mode. The XRD patterns have been supplemented in the revised Supporting Information (Figure S6).

Figure R2. Powder XRD patterns of COF-V-x powders (x=30%, 60% and 80%). The AA stacking mode is included as a reference.

Revised manuscript

Figure 2. Preparation and characterization of COF-Cys membranes.

Revised Supporting Information

Figure S6. Powder XRD patterns of COF-V-x powder samples (x=30%, 60% and 80%).

5. In the caption of Figure 2, parentheses are used in part of the letters, but not in other places in the manuscript. It is recommended to revise them to keep consistent.

Response: Thanks for the reviewer's reminder. The parentheses in Figures 2d, e, and f have been removed in the revised manuscript.

Revised manuscript

Figure 2. Preparation and characterization of COF-Cys membranes. **a.** Interfacial polymerization of COF-V-x membranes and rapid synthesis of COF-Cys-x membranes via a thiol-ene click reaction. **b.** SEM image of COF-Cys-60% membrane showing defect-free surface morphology (scale bar, 500 nm). Projection of the in-plane diffraction patterns from the grazing incidence wide-angle X-ray scattering (GIWAXS) data: **c.** COF-V-x membranes, **d.** COF-Cys-x

membranes on PAN support ($x=30\%$, 60%, and 80%). Insets show GIWAXS patterns. The PXRD pattern of COF-V-60% powder is included as a reference. Low-dose motion-corrected high-resolution TEM (HRTEM) images: **e.** COF-V-60% membrane (scale bar, 50 nm), **f.** COF-Cys-60% membrane (scale bar, 50 nm), suggesting high crystallinity.

Reviewers' Comments:

Reviewer #1:

Remarks to the Author:

The authors have well addressed my concerns regarding the experimental details and MD simulations. Debatable descriptions have been corrected. The revised manuscript can be accepted for publication.

Reviewer #2:

Remarks to the Author:

In the revised manuscript, the authors have answered in detail all questions raised by the referees. Therefore, I recommend the paper for publication without the further revision.

A point-by-point response to referees' comments

Reviewer #1 (Remarks to the Author):

The authors have well addressed my concerns regarding the experimental details and MD simulations. Debatable descriptions have been corrected. The revised manuscript can be accepted for publication.

Response: We greatly appreciate the reviewer for the highly positive comment.

Reviewer #2 (Remarks to the Author):

In the revised manuscript, the authors have answered in detail all questions raised by the referees. Therefore, I recommend the paper for publication without the further revision.

Response: We greatly appreciate the reviewer for the highly positive comment.